# The Impact of Social Distancing Due to the COVID-19 Pandemic on People with Dementia, Family Carers and Healthcare Professionals: A Qualitative Study

**DOI:** 10.3390/ijerph19010519

**Published:** 2022-01-04

**Authors:** Hanneke J. A. Smaling, Bram Tilburgs, Wilco P. Achterberg, Mandy Visser

**Affiliations:** 1Department of Public Health and Primary Care, Leiden University Medical Center, P.O. Box 9600, 2300 RC Leiden, The Netherlands; H.J.A.Smaling@lumc.nl (H.J.A.S.); w.p.achterberg@lumc.nl (W.P.A.); 2Department of Intensive Care Medicine, Radboud University Medical Center, Radboud Institute for Health Science, P.O. Box 9101, 6500 HB Nijmegen, The Netherlands; Bram.Tilburgs@radboudumc.nl

**Keywords:** dementia, COVID-19, social isolation, long-term care, home care, family carers

## Abstract

Social distancing measures imposed because of the COVID-19 pandemic presented challenges to the health and wellbeing of people with dementia, family carers, and healthcare professionals. This study investigated the impact of these measures on all involved in the care for people with dementia. For this qualitative study, 20 family carers and 20 healthcare professionals from home care and long-term care (LTC) participated in a semi-structured interview. Interviews were analysed using an inductive thematic analysis approach. For people with dementia, the social distancing measures resulted in a deterioration of physical health. The impact on their emotional state and behaviour depended on the stage of dementia. Family carers experienced difficulty coping with visiting restrictions, anxiety regarding safety, and changes in carer burden. Healthcare professionals had an increased workload, and felt guilty about adhering to restrictive measures. Differences between home care and LTC were reported (i.e., societal initiatives focussed on LTC, scarcity of activities for community-dwelling people with dementia, use of personal protective equipment more intrusive for home care). The social distancing measures had a negative impact on persons with dementia, their family carers, and healthcare professionals. More attention is needed for community-dwelling people with dementia and family carers in times of social isolation.

## 1. Background

In 2020, nearly 80% of deaths due to COVID-19 concerned people aged 65 and older with multiple chronic conditions [1]. Nursing home residents, on average, made up 46% of COVID-19 related deaths [2]. To protect vulnerable groups and limit community spread, governments have been imposing social distancing measures since early 2020. In the Netherlands, social contact outside the household was initially allowed on the conditions that it was restricted to a maximum of three persons and at least 1.5 m distance. Later in 2020, this was further reduced to one person [3]. A partial lockdown started in October 2020 and was followed by the introduction of an evening curfew in January 2021 [4]. 

In the first week of March 2020, most mental health and nursing home organizations took the first measures to prevent the risk of infection for their patients and staff. Visitors were banned from nursing homes between mid-March and mid-May 2020 [3]. Additionally, residents were prevented from going outside and social and group activities were halted. Where possible, healthcare professionals (HCP) were asked to work from home to minimize staff infection. As a result, all group sessions and activities, and outpatient care that was not considered urgent were cancelled [5]. For an elaborate overview of the preventives measure taken by the Dutch government, see [4].

These measures presented vital challenges to the health and wellbeing of residents of long-term care facilities (LTCF) and community-dwelling older adults, particularly those who are frail or have multiple chronic conditions. 

The physical health and wellbeing of people with dementia is negatively impacted by disruptions in their daily routines. When facing the challenges of social distancing measures, they are particularly susceptible to rapid deterioration. Restricting movements for people with dementia and their carers represents a significant loss of autonomy, with psychological and physical harms associated with social isolation and immobility [6]. Additionally, dementia can worsen under stress and as a result of changes in daily routines. The social distancing measures, often accompanied by the suspension of non-essential care in both home care and long-term care (LTC), and the decision to ban visitors from LTCF conflicted with goals of care [7,8] and may have intensified loneliness and anxiety people with dementia and their family carers [9,10] resulting in social isolation. Withdrawal of the formal and informal support on which many people with dementia and their family carers rely, may compound these problems, especially within the community setting.

Family carers provide a substantial amount of care for their relative with dementia [11]. Providing care for a person with dementia is generally considered burdensome [12] and related to an increased risk of physical and psychosocial health problems [12,13,14]. Carer burden does not automatically decrease after LTC admission [15]; family carers often stay involved [16,17]. Many family carers who were providing unpaid, essential care to LTC residents before the pandemic were now unable to, which likely contributed to reported increases in staff workload, stress and burnout [7,18]. Furthermore, even before the pandemic, more people with dementia were staying at home longer. This may be increased during the pandemic due to admission stops [19,20], fear for not being able to visit their relative due to visitor bans, and because LTCF were considered the most important hotspot for COVID-19 deaths [19,21]. Additionally, home care restrictions asked even more from family carers. This makes gaining insight into the experiences and needs of family carers and HCP during the COVID-19 pandemic highly relevant, and may aid the support of family carers and HCP to mitigate negative health consequences for all those involved [12].

Many LTCF have made efforts to keep residents socially engaged, for example, through window visits and video chats. Although helpful for some, these interactions may be confusing and frustrating for people with dementia. Additionally, seeing family may not have the same beneficial impact as physical forms of affection and communication. Hearing and vision loss can make communicating during a physically distanced visit challenging, and the use of personal protective equipment (PPE) by family carers HCP may contributed to further distress in people with dementia. The impact of these alternative ways of social contact on people with dementia, their family carers and HCP have not been studied yet.

The social distancing measures, with the lack of social interaction and physical closeness, are thought to have an adverse impact on the well-being of on people with dementia and their carers [22,23,24,25,26]. The first studies focussing on the effects of the restrictive measures are now emerging, substantiating these claims [22,27,28]. Challenging behaviour seemed to increase in people with dementia, mainly triggered by extended isolation [27,28]. Family carers experienced psychological problems, and felt less supported when they had to handle challenging behaviour or offer meaningful activities to their relative [29]. Less attention has been paid to the needs of people with dementia, their family carers and HCP during times of limited social contact, especially for community-dwelling people with dementia. 

To our knowledge, to date no study has investigated the impact of the social distancing measures during the COVID-19 pandemic on the social wellbeing of all those involved (e.g., people with dementia, family carers and HCP) in both home care and LTC. Knowing their needs and the impact of the social distancing measures helps home care services, LTCF, policymakers, and governments optimize dementia care in situations that may require social distancing measures. By including both home care and LTC, important differences (and similarities) in needs and impact can be identified enabling us to further optimize support and care. This study investigates the impact of social distancing measures during the COVID-19 pandemic on home care and LTC for people with dementia, family carers and HCP.

## 2. Methods

### 2.1. Design

This interview study has a generic qualitative design. It investigates people’s reports of their subjective opinions, attitudes, beliefs, and reflections on their experiences of things in the outside world [30]. Generic qualitative studies do not claim allegiance to a single established methodology, while main principles such as the constant comparative method are borrowed from traditional qualitative approaches [31]. Semi-structured interviews were conducted with 20 family carers and 20 HCP of people living with dementia. 

The study was exempt from the Medical Research Involving Human Subjects Act by the Medical Ethics Review Committee Leiden-The Hague-Delft. (protocol number: CoCo 2020-067). All participants provided written informed consent. This study was registered at the Long-Term Care response to COVID-19 website on 10 February 2021 (https://ltccovid.org/project/covid-19-and-social-isolation-in-dementia-care-isolate-impact-and-needs-of-people-with-dementia-informal-and-professional-caregivers/).

### 2.2. Setting

In the Netherlands, around 280,000 people were living with dementia in 2019, with 29% of them residing in LTCF [32]. During the onset and early stages of dementia, support is mostly provided by primary care practitioners, family carers and patient organizations. After diagnosis, local services determine specific home care packages such as case management, support groups, respite care or counselling. When living at home is no longer possible, it is customary for people with dementia to relocate to a LTCF (i.e., nursing home or residential care/assisted living) [33]. The provision of care in LTCF mostly relies on non-profit providers [34]. Nursing homes provide 24-h care and oversight by multidisciplinary teams. The residents require assistance with activities of daily living and often have complex health needs [35]. More than 60% of the nursing home residents are females of 80+ years and 85% of the residents have more than two chronic conditions [36]. The ‘first wave’ of COVID-19 infections in the Netherlands was defined based on excess mortality from week 11 to 19 2020, and a ‘second wave’ from week 39 2020 to week 24 2021 [19].

### 2.3. Participants

The inclusion criteria were (1a) being a family carer of a person with dementia who received home care or resided in a LTCF during the COVID-19 pandemic, or (1b) being an HCP working at a home care service or LTCF who provided care to people with dementia during the COVID-19 pandemic (March 2020 to March 2021), (2) having sufficient understanding of the Dutch language to participate in an interview, and (3) being >18 years. Participants were excluded if they indicated any mental or cognitive problems.

Participants were recruited via the Dutch Alzheimer’s Association by a one-time request by email, resulting in 51 family carers and 13 HCP wanting to participate. HCP were also recruited via the University Network for the Care sector South-Holland (n = 12). To ensure diversity in our sample, participants were selected based on the setting (community vs. LTC), and gender, and specifically for family carers; relationship to care recipient, and type of dementia. Of the 20 initially selected family carers, 4 dropped out before signing the informed consent due to personal reasons (n = 1) or without stating a reason (n = 3). Of the HCP, 5 dropped out due to personal reasons (n = 2), being too busy (n = 1), or without mentioning a reason (n = 2). These dropouts were replaced with other participants who had previously expressed their interest to participate in the study. 

### 2.4. Data Collection

Semi-structured interviews were conducted with 10 family carers and 10 HCP caring for people with dementia at home and 10 family carers and 10 HCP from LTC. The interview guide is presented in Appendix A. The interviews were conducted by three postdoctoral dementia researchers (one male (BT), two females (MV, HS); all with ample experience with conducting interviews with vulnerable populations), and two female medical students (EvdB, BvL). The medical students received a short interview training and received elaborate feedback on their first two interviews by two researchers (HS, MV). Due to the COVID-19 restrictive measures, the interviews took place online via the software application ZOOM. The interviews were audio recorded and transcribed verbatim. The participants were asked if they wanted to receive a copy of the transcript. The data were collected between November 2020 and March 2021. A sample size of 20 participants per setting was deemed sufficient to reach data saturation [37,38].

### 2.5. Analysis

Inductive thematic analysis was used, meaning the data itself were used to derive the structure of the analysis and no predetermined theory or framework was used [39]. An initial list of codes was developed from the first three transcripts, independently by two researchers (HS, EvdB). A consensus list of codes was created. This was used to recode the first three interviews. Next, the consensus list was discussed within the team and applied to the analysis of the next three transcripts. When new concepts emerged from the analysis of the second batch, they were added to a revised coding framework. The previous transcripts were then recoded using the revised framework, and the process was repeated through all the transcripts, with the coding being repeatedly revised. 

To ensure inter-rater agreement, two researchers (HS, MV) did the analysis; similar and related codes were organized into higher level categories, which were restructured into a hierarchy of more abstract, overarching thematic clusters and sub-themes to form a coherent whole. The researchers engaged in a reiterant process of discussing areas of agreement and disagreement to enhance analytical rigor and achieve consensus. The research team was consulted if consensus could not be achieved. The outcomes were discussed within the research team and the other researchers provided feedback on the analysis. Qualitative analysis software ATLAS.ti (ATLAS.ti Scientific Software Development GmbH, Berlin, Germany) was used to facilitate the analysis process. 

## 3. Results

The average age of the 40 interviewees was 58.4 years (SD 15.7, range 20–84). In total, 60% were female (n = 24). All of the 24 participants whose ethnicity was known, were from the Netherlands. The family carers were either the partner (65%, n = 13) or daughter of the person with dementia (35%, n = 7). Of the family carers of home-dwelling persons with dementia, the partners lived with the care recipient, two daughters provided daily care together with the partner of the care recipient, and one provided care together with her siblings to her parent who was living independently in an assisted living apartment for seniors. The interviews lasted on average 58.7 min (SD 8.4, range 43–83). Table 1 presents the characteristics of the interviewees per group and per setting.

The qualitative data on the impact of the social distancing measures are grouped in four themes: (1) impact on people with dementia (physical, cognitive, behavioural and emotional changes), (2) impact on family carers (communication alternatives, anxiety, carer burden), (3) impact on HCP (workload, adhering to measures, emotional impact), and (4) differences between home care and LTC (societal, daily life, use of PPE).

### 3.1. Theme 1. Impact on Person with Dementia

According to HCP, the decreased opportunities to go outside and reduced physical activities often led to a deterioration of physical health in people with dementia (e.g., weight gain, muscle tightness). In some cases, community-dwelling people with dementia experienced significant weight loss when home care services were downscaled during the first wave of the COVID-19 pandemic, as explained by an HCP:
*To reheat it [the meal] together with the client and to make sure that the client would actually start eating. Well, we scaled that down, because at that moment the plan was, we’ll telephone the client about it and we’ll make sure everything’s ready to go. Well, this client lost four kilos in about two weeks.*(HCP 29, home care)

Interviewees were conflicted about whether the social distancing measures, including restrictions of social contacts and activities, affected people’s cognitive condition in any way: both family carers and HCP noticed cognitive decline in people with dementia, but also acknowledged that these changes could have been caused by the natural progression of the disease. 

Often mentioned behavioural changes in people with dementia in LTC were increased restlessness and aggression. In the community setting, social distancing measures led to depression and boredom. As one HCP in home care explained:
*Of course, they’re people, and they have been done with their life for a while. And they are kind of waiting, you know? I have a gentleman, 93 years old, and he says, well it really isn’t much fun anymore. I pray every night that I won’t open my eyes again. So this gentleman simply says give me the coronavirus.*(HCP 26, home care)

Interestingly, the perceived impact of the social distancing measures on the emotional state of the persons with dementia depended on the stage of dementia and the care setting, according to both family carers and HCP. On the one hand, in LTC, people with mild to moderate dementia were emotional (e.g., lonely, angry) due to the lack of contact with their relatives, whereas people with severe dementia in LTC were less affected by the social distancing measures. They were mostly unaware of the visiting restrictions in LTCF, according to the interviewees. On the other hand, in the community, people with mild dementia were coping relatively well. People with more severe dementia were often challenged in many ways due to a lack of additional home care services and community support, which resulted in negative emotions (e.g., anger, aggression). So, in LTC, people with mild to moderate dementia were more emotionally affected than people with severe dementia by the social distancing measures, whilst in home care, people with mild to moderate dementia were less emotionally affected than people with severe dementia. 

### 3.2. Theme 2. Impact on Family Carer

Due to the social distancing measures, family carers were often unable to visit their relatives in LTC. Both family carers and HCP explained that the alternative forms of communication, such as video calling and window conversations, were not ideal, since people with dementia often had difficulties using these communication aids. However, family carers stated it was better than no communication at all. The majority of interviewees mentioned that the visiting restrictions were difficult for relatives to cope with, as they were scared their relatives with dementia would not recognize them once the restrictions were lifted. In the community, the ‘bubble’ around the person with dementia was kept small. It was mostly limited to visits from the primary family carer(s) if the person with dementia was living by themselves or only a handful of other family carers if the person with dementia lived with their partner.

Although family carers understood that visiting restrictions and PPE regulations in home care and LTC were meant to keep their relative safe, several family carers were afraid HCP would not adhere to the general safety measures. One family carer explained:
*That’s why I was so incredibly shocked that people from the nursing home didn’t wear the protection that I had to wear at the end of May. While they just walked around and picked up their kids from school and went everywhere, while I thought, well. I’m on Zoom all day for work, I don’t have kids in the house, I don’t have to go to school, I do my shopping online or whatever. I’m at less risk. So it’s weird that I should have to adhere to different measures, that different measures are applied to me than to the people at the nursing home. That’s just not right.*(Family carer 7, LTC)

In general, family carers and HCP indicated the carer burden of family carers of nursing home residents decreased, as opposed to those who cared for community-dwelling people with dementia. Interviewed family carers of nursing home residents felt side-lined as they had to hand over their last caregiving responsibilities:
*Look, rationally, I thought, yes, this is the best solution. Emotionally it’s something completely different of course, because I’d already had to let go once. From 24 h caring a day to 4 h caring a day. That was also very difficult, to leave that to others. And now, at a certain point, nothing.*(Family carer 10, LTC)

Family carers of community-dwelling people with dementia explained that care routines often intensified:
*My most distressing experience is of a man who sort of fell through the cracks. Because this man was actually ready for day care... he wasn’t in day care yet, but he should have been. But because he wasn’t, I had nothing to offer him. I couldn’t offer a volunteer because they didn’t come in during the pandemic. I could not offer any day care at that time, because there was none. I could not offer individual counselling, because I was not allowed to. And his partner, she called us almost every day crying that it was too much for her.*(HCP 25, home care)

However, despite an increase in care responsibilities, several family carers of community-dwelling people with dementia underlined the COVID-19 pandemic did not make them feel more isolated than usual:
*I feel isolated, but that has little to do with COVID. It’s just that I can’t live my own life and I have to take someone else into account all the time, and I do, but sometimes it’s a bit much.*(Family carer 31, home care)

### 3.3. Theme 3. Impact on HCP

In both home care and LTC, HCP’s workload increased due to the additional responsibilities and care tasks they were given. For example, when a ward or individual resident needed to be quarantined, being more vigilant for COVID-19 symptoms, the use of PPE increased, and caring for COVID-19 patients intensified. HCP often had to work extra hours because of staff shortage due to sick leave. HCP in home care also indicated that it was burdensome having to decided what was essential care and having to communicate to patients when (part of) their home care had to be temporarily stopped.

According to HCP in home care, after the first wave of the COVID-19 pandemic in June 2020, it became clear that community-dwelling people with dementia were in greater need of extended home care services. As a result, instead of scaling down essential care as much as possible, HCP explained they were expected to work extra shifts and longer hours, to cover for their colleagues on sick leave or to provide extra care for those in need:
*And at some point there was this shift, when obviously we stopped scaling down and started to scale up. And then came pressure from the hospitals. Because patients had to go home. And you needed to deal with that too. So we looked at each other and we said, we’ll just have to pull together.*(HCP 29, home care)

Moreover, HCP indicated that their work entailed stressful situations, as relatives of people with dementia did not always adhere to rules imposed by LTCF. One HCP outlined how she had to call security and sometimes even emergency services to keep family members off the premises. Interviewed HCP were generally understanding towards this type of behaviour, despite the negative effect of these stressful situations on their work experience. They found some measures too strict, and in some cases they regretted having enforced the rules. One HCP working in a LTC explained:
*The restrictions on family allowed to be with their dying father or mother, in a protective suit, at 1.5 metres, 3 visitors per 24 h. So one visitor allowed per 8 h. Then they were not allowed to use anything. You couldn’t offer them coffee, because they were not allowed to remove the protection, and then the person had died and they were quickly taken away in double body bags. We couldn’t see them off. We couldn’t do anything at all. And that was it. Well, that’s just not the kind of care you want to provide. And informal carers, wives who first haven’t seen their husbands for a long time. Eventually he dies. And then you have to say, well, don’t touch him. That was just unbearable. But you had no choice. That really got to me.*(HCP 15, LTC)

Some HCP admitted they felt uncomfortable, almost guilty, that they were able to have contact with residents and family members were not:
*This was very difficult, because I would put an arm around their father or mother and they’d be standing there. There were parties, birthdays were celebrated in that way. One gentleman died and that was his last birthday. There’s a clip where I’m dancing with a few residents there and then you see that family at a distance. Yes, that’s simply heart-breaking.*(HCP 18, LTC)

### 3.4. Theme 4. Differences between Home Care and LTC 

Several interviewees underlined that since the outbreak of the COVID-19 pandemic, the Dutch community has shown great public efforts to support people with dementia in LTC. For example, one interviewee said:
*Restaurants, huge amounts of food, because they had supplies they couldn’t use anymore. Yes, heart-warming. Or neighbourhood children who had all made drawings and sent cards. And yes, it is really fantastic to see what the Netherlands is like then. So, so kind.*(HCP 14, LTC)

Interviewed HCP mentioned that creative initiatives emerged within the community to hearten nursing home residents, such as so-called window-performances by musicians or outdoor bingo games. However, family carers and HCP also acknowledged there was less societal concern and effort for community-dwelling people with dementia.

Several family carers and HCP mentioned there was a scarcity of activities organized for community-dwelling people with dementia. Especially during the first wave of the COVID-19 pandemic (March until June 2020), respite care and day care centres shut down without providing any alternative services. According to both family carers and HCP, this resulted in a double dichotomy: not only did the lives of people at home change drastically compared to the lives of people in LTC, but community-dwelling people with dementia with a limited social network were also substantially more affected by the COVID-19 pandemic than those who had a large network to rely upon. As an HCP working in home care explained:
*I have a client with dementia who has no other relatives. Because you see that sons and daughters of clients with dementia continue to visit. Sometimes a little less frequently, but they still visit. But this one client has no family. She has to rely on her neighbours for social contact. And you notice that they are quicker to say, well, sorry, but I’m not coming then.*(HCP 25, home care)

The use of PPE was less intrusive for people involved in LTC compared to home care, according to interviewees. In LTC, residents were familiar with PPE as HCP and visitors were obliged to use PPE non-stop, opposed to community-dwelling people with dementia, who were only confronted with PPE by HCP visiting them at home. Face masks emphasized the apparent social distance between the HCP and their home care clients:
*Especially with people with poor hearing, I notice that you communicate less, maybe a little more detached, because of the face mask. Of course, there is the physical barrier of the mask, so they can’t read your lips. And that, of course, helps them understand others. And it also creates a kind of extra, semi-professional distance.*(HCP 23, home care)

HCP working in home care explained that using PPE was time consuming, as they had to disinfect and change PPE after every home visit.

## 4. Discussion

This study investigated how social distancing measures during the COVID-19 pandemic impacted home care and LTC for people with dementia, family carers and HCP. For people with dementia, the impact was seen in a deterioration of physical health and behavioural changes. The impact on their emotional state depended on the stage of dementia, while the impact on their cognitive status remains unclear. Family carers struggled to cope with visiting restrictions, anxiety about their relative’s safety, and change in carer burden. HCP had an increased workload, stress and regretted to adhere to the restrictive measures, and feelings of guilt. Community-dwelling people with dementia with a limited social network were particularly affected by social distancing measures. 

This study underlines the importance of monitoring physical and emotional wellbeing of people with dementia in times of social isolation, and responding accordingly. Our results show a major negative impact of social distancing measures on people with dementia’s physical health, their emotional state and behaviour, which ties in with earlier studies on social isolation [27,28,40,41]. Restrictions of physical contact with people outside of nursing homes endanger the mental health of residents [21,42]. There is now international consensus that nursing homes need to keep open for visitors safely with appropriate consideration of community transmission, nursing home outbreak status, and the preferences of residents, families, and HCP [7].

Our observed detrimental effects on emotional wellbeing were more pronounced for community-dwelling people with more severe dementia, and nursing homes residents with mild to moderate dementia. In line with earlier work, this study showed that the loss of purpose stemming from meaningful activities contributed to mental problems, at least for community-dwelling people with dementia [43]. In order to preserve and protect the emotional wellbeing of people with dementia, social distancing measures should be replaced by provisions allowing social contact without significantly increasing risks of infections, and meaningful activities should be offered as much as possible, also in the community setting. Safe visiting practices as suggested by earlier research include visits in well ventilated separated areas or outdoors, proper infection control procedures, registration and screening of visitors, limiting the number, frequency and length of visits, and supervised by staff [7].

Because of the lengthy duration of the COVID-19 pandemic, it was unclear for the interviewees whether any cognitive deterioration was caused by understimulation due to the social distancing measures or was part of the disease progression. As people with dementia themselves worry about a faster cognitive decline [40] and experience a decline in their cognitive abilities due to the limited social contacts [43], more (longitudinal) research on this topic is necessary—not only to determine the impact of social distancing measures on the cognitive status of people with dementia, but also to develop possible interventions to limit the detrimental effects of social isolation.

The visiting restrictions in LTC seemed particularly distressing for family carers. Communication through electronic devices was not always a suitable alternative, as most residents had difficulty using them. Video-conferencing has proven to be challenging, even for older adults in LTC without dementia [44]. In general, we want to encourage family carers and HCP to practice physical distancing while staying socially engaged with people with dementia, as meaningful interactions and activities are vital for quality of life and remaining healthy. More effort could be made towards alternative ways of social contact, such as better facilitated distanced visiting, or finding other ways to keep family carers involved. Clear reasoning and communication with people with dementia, family carers and HCP about how social distancing measures are developed is vital, as is flexibility in enforcing those measures [7,8].

Almost all family carers of community-dwelling people with dementia in our sample were either living with the person with dementia or providing daily. These family carers were highly affected by home care services scaling down, especially when they cared for someone with moderate to severe dementia. The expansion of the responsibility of care for family carers is also observed for home care recipients in general [45]. Interestingly, although a substantial number of HCP voiced their concerns about the wellbeing of those family carers, family carers themselves often experienced that their voices were not heard anyway, and they were already isolated, regardless of the pandemic and its restrictive measures. Support for family carers of people with dementia, especially during social isolation is limited, and should be intensified [40,46].

HCP have been challenged in many ways during the COVID-19 pandemic: apart from keeping themselves and their clients safe from infections, their workload increased and required them to be more flexible. They encountered challenging behaviour on the part of their clients and family carers. Pro-actively supporting the wellbeing of HCP is needed, as they have been exposed to high-stress environments, which may cause mental health issues [6,18].

This study showed how social distancing measurements affected people with dementia, their family carers and HCP in home care and LTC. Unfortunately, we were unable to include people with dementia as participants. Both the reflective approach of the interviews and the restrictions around physical visits meant that online interviews on experiences during the course of the pandemic were not deemed feasible for people in the more advance stages of dementia. Additionally, we only had data of less than half of family carers from the community setting about how the hours they spend on providing care for their relative changed due to the pandemic. However, this study used a relatively large sample size to amplify the voices of family carers and HCP of people with dementia in both home care and LTC, and is therefore a valuable contribution to the growing body of research on implications of the COVID-19 pandemic in dementia care.

## 5. Conclusions

At the outbreak of the COVID-19 pandemic, dementia care policies had to be developed swiftly, and were most applicable to LTC residents. However, this study showed that the needs of those involved may vary and that home care services required different approaches than LTC. More care during times of social distancing measures is needed for community-dwelling people with dementia, especially those with a limited social network, or those in need of intensified care for other reasons. In responding to the pandemic, it is essential to be mindful of the challenges that restrictive measures are creating for people with dementia and their family carers and to address these challenges head on. Therefore, we advise family carers and HCP to follow social distancing measurements while staying socially engaged with people with dementia. Additionally, we recommend to involve persons with dementia, their advocates, families, and HCP in discussions around restrictive measures, in particular regarding safe visiting.

## Figures and Tables

**Table 1 ijerph-19-00519-t001:** Sample characteristics per setting.

	Home Care Setting			Long-Term Care Setting		
	n	M [SD]/%	Range	n	M [SD]/%	Range
Family carers						
Age (years)	9	65.3 [16.4]	30–81	9	66.1 [13.1]	43–84
% female	4	40%		6	60%	
Relation to person with dementia	10			10		
− Partner − Child	7 3	70% 30%		6 4	60% 40%	
Educational level	10			10		
− Secondary vocational education − Bachelor’s or Master’s degree	3 7	30% 70%		1 9	10% 90%	
Type of dementia relative	8			8		
− Alzheimer’s dementia − Other	7 1	87% 13%		5 3	63% 37%	3–30
Time spent on caregiving tasks (hours a week)	9	90.8 [71.5]	7–168	10	13.5 [9.4]	
Intensity of care compared to before pandemic	4			7		
− Less − Even − More	1 2 1	25% 50% 25%		4 0 3	57% 0% 43%	
Healthcare professionals						20–66
Age (years)	8	53.4 [9.2]	34–64	10	49.4 [16.7]	
% female	6	60%		8	80%	
Educational level	9					
− Secondary vocational education − Bachelor’s or Master’s degree	2 7	22% 78%		9 1	90% 10%	1–45
Experience as a healthcare professional (years)	9	16.3 [16.2]	3–43	8	19.6 [17.1]	4–70
Time spent on caregiving tasks (hours a week)	10	35.1 [9.6]	15–50	10	30.5 [16.8]	
Intensity of care compared to before pandemic	7			6		
− Less − Even − More	0 4 3	0% 57% 43%		0 1 5	0% 17% 85%	

## Data Availability

The data are not publicly available due to restrictions (e.g., privacy and limitations in consent about sharing data with others).

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
