# Peer review of "The Impact of Social Distancing Due to the COVID-19 Pandemic on People with Dementia, Family Carers and Healthcare Professionals: A Qualitative Study"

_ijerph, 2022, doi:10.3390/ijerph19010519_

Round 1

Reviewer 1 Report

Thank you for letting me review this interesting paper, the paper is coherent to the scope of the journal.

I suggest to edit the references according to the instruction for authors of the journal.

Author Response

Reviewer report 1

  1. I suggest to edit the references according to the instruction for authors of the journal.

We would like to thank the reviewer for their suggestion. We have downloaded the journal’s suggested EndNote style and used this format for our references.  

Reviewer 2 Report

In this manuscript, the authors investigated how the social isolation measures affected the patients with dementia and their family carers and healthcare professionals in both home care and LTC during COVID-19 pandemic. This is a very important study that gave us insight into how the social isolation impacted the physical health and the cognitive status of people with dementia.  The study is well written with a standard English language. I have just two questions for the authors:

  1. Regarding the data for patients at home care, were the participants from the same home care?
  2. Have the authors considered to conduct a control group? I can understand how difficult to do so but I was just wondering.

Author Response

Reviewer report 2

In this manuscript, the authors investigated how the social isolation measures affected the patients with dementia and their family carers and healthcare professionals in both home care and LTC during COVID-19 pandemic. This is a very important study that gave us insight into how the social isolation impacted the physical health and the cognitive status of people with dementia. The study is well written with a standard English language. I have just two questions for the authors:

  1. Regarding the data for patients at home care, were the participants from the same home care?

All the family carers in our study were recruited via Alzheimer Nederland, the Dutch Alzheimer’s Association. This ensured that we covered all of the Netherlands and that participants receiving home care were from various home care services.

  1. Have the authors considered to conduct a control group? I can understand how difficult to do so but I was just wondering.

We would like to thank the reviewer for their suggestion. Unfortunately, it was impossible to include a control group, as the COVID-19 pandemic is a world-wide problem. The Netherlands is a relatively small country. The restrictive measures from our government were similar throughout our country, making it impossible to find a comparable control group that was not effected by COVID-19 and/or the national imposed restrictive measures.

Reviewer 3 Report

The impact of COVID-19 does not only include “social isolation.” In title, you emphasized “social isolation.” However, in your abstract purpose, This study investigated the impact of these measures on all involved in the care for people with dementia. Are preventive measures the focus of the study or the social isolation (results of preventive measures) being the focus? I feel that preventive measures toward COVID-19 can cause social isolation, but the measures are not meant to cause social isolation. It is a bit odd to call the measures “social isolation measures.” Please consider and be clear. It seems to me the measures that you mentioned in the manuscript include change of the ways of healthcare practice. For example, some services are replaced with distant services, the distance services are not meant for social isolation, they are actually services to counter social isolation.

The abstract conclusion seems not based on the results presented. The implications “clear communication…,” but your themes and categories does not seem to relate to communication problems.

Introduction

Please be more specific regarding the background information of changes in practice in home care and LTC settings. What services were provided? What services were banned? What were the changes of the services/measures? The link among preventive measures, impact of preventive measures, change of services, and social isolation could be clearer.

Method

  1. Please consider describing preventive measures and practice change in the settings so that when you explore the impact of the measures, the international readers can follow.
  2. Participants were recruited via [BLINDED FOR REVIEW] by a one-time request by email. What was the population list? How did you consider the characteristics of the replacement sample?
  3. You applied several interviewers. What training did they have?
  4. How do you ensure results rigor?

Results

  1. I felt that family caregivers and health workers should be separated throughout. They are two different group of people.
  2. The four themes: 1) impact on people with dementia, 2) impact on family carers, 3) impact on HCP, and 4) differences between home care and LTC could be renamed to reflect what they are. Now it looks like four aspects of enquiry, not themes. Were there any similar impacts for the 3 groups and 2 settings?
  3. Table 1. Do they have increased or decreased care hours during the pandemic period?
  4. You mentioned “In both home care and LTC, HCP’s workload increased…” As you mentioned earlier, many services were shut down. Why all workers’ workload increased? How about work hours?

Discussion

  1. Please be more specific about practice suggestions. Is there any better way than restrictive measures? How can we do better? Please distinguish what has been done what has not been done.
  2. People staying home much longer in lockdown. If patients with dementia lived with family members, did they spend more time or have more interaction with them?
  3. Conclusion could be modified to be more related to your results.

Author Response

Reviewer report 3

The impact of COVID-19 does not only include “social isolation.” In title, you emphasized “social isolation.” However, in your abstract purpose, This study investigated the impact of these measures on all involved in the care for people with dementia. Are preventive measures the focus of the study or the social isolation (results of preventive measures) being the focus? I feel that preventive measures toward COVID-19 can cause social isolation, but the measures are not meant to cause social isolation. It is a bit odd to call the measures “social isolation measures.” Please consider and be clear. It seems to me the measures that you mentioned in the manuscript include change of the ways of healthcare practice. For example, some services are replaced with distant services, the distance services are not meant for social isolation, they are actually services to counter social isolation.

We agree with the reviewer that we could have been more precise with the terminology. We agree that the measures were not taken to cause social isolation, but may have resulted in social isolation. To avoid further confusion, we have replaced social isolation (measures) by social distancing measures where appropriate.

The abstract conclusion seems not based on the results presented. The implications “clear communication…,” but your themes and categories does not seem to relate to communication problems.

We like to thank the reviewer for their comment. The conclusion of the Abstract on p2 has been adjusted to: “The social distancing measures had a negative impact on persons with dementia, their family carers, and healthcare professionals. More attention is needed for community-dwelling people with dementia and family carers in times of social isolation.”

Introduction

Please be more specific regarding the background information of changes in practice in home care and LTC settings. What services were provided? What services were banned? What were the changes of the services/measures? The link among preventive measures, impact of preventive measures, change of services, and social isolation could be clearer.

We have added more information to the Introduction about the preventive measures and practice changes in the home care and LTC setting in the first two paragraphs. They now read:

“In 2020, nearly 80% of deaths due to COVID-19 concerned people aged 65 and older with multiple chronic conditions[1]. Nursing home residents, on average, made up 46% of COVID-19 related deaths[2]. To protect vulnerable groups and limit community spread, governments have been imposing social distancing measures since early 2020. In the Netherlands, social contact outside the household was initially allowed on the conditions that it was restricted to a maximum of three persons and at least 1.5 metres distance. Later in 2020 this was further reduced to one person[3]. A partial lockdown started in October 2020 and was followed by the introduction of an evening curfew in January 2021[4].

In the first week of March 2020, most mental health and nursing home organizations took the first measures to prevent the risk of infection for their patients and staff. Visitors were banned from nursing homes between mid-March and mid-May 2020[3]. Also, residents were prevented from going outside and social and group activities were halted. Where possible, healthcare professionals (HCP) were asked to work from home to minimize staff infection. As a result, all group sessions and activities, and outpatient care that was not considered urgent were cancelled[5]. For an elaborate overview of the preventives measure taken by the Dutch government, see [4]. These measures presented vital challenges to the health and wellbeing of residents of long-term care facilities (LTCF) and community-dwelling older adults, particularly those who are frail or have multiple chronic conditions.”

An additional sentence has been added to p4: “The social distancing measures, with the lack of social interaction and physical closeness, are thought to have an adverse impact on the well-being of on people with dementia and their carers [22-26]”

We are confident that with the additional information the link among preventive measures, impact of preventive measures, change of services, and social isolation is now more clear.

Method

1. Please consider describing preventive measures and practice change in the settings so that when you explore the impact of the measures, the international readers can follow.

Please see our previous response for the additions we made to the Introduction regarding describing the preventive measures and practice changes in the settings.

Under Setting, Methods, we have added more information about the 1st and 2nd COVID-19 waves in the Netherlands for the international readers:

“The ‘first wave’ of COVID-19 infections in the Netherlands was defined based on excess mortality from week 11 to 19 2020, and a ‘second wave’ from week 39 2020 to week 24 2021 [19].” p5.

2. Participants were recruited via [BLINDED FOR REVIEW] by a one-time request by email. What was the population list? How did you consider the characteristics of the replacement sample?

An one-time request was sent to the panel of the Dutch Alzheimer’s Association. This ensured that we covered all of the Netherlands. This panel consists of persons with dementia, family carers, and HCP from diverse backgrounds who are willing to share their experiences and opinion about all things dementia and caregiving related. The replacement sample was selected based on the same criteria as the initial sample. To clarify in the manuscript, we made the following adjustment on p5;

“To ensure diversity in our sample, participants were selected based on the setting (community vs. LTC), and gender, and specifically for family carers, relationship to care recipient, and type of dementia”

3. You applied several interviewers. What training did they have?

The three postdocs all had ample experience with interviewing people from vulnerable populations. Therefore additional training was not deemed necessary. The medical students received a short interview training and got elaborate feedback on their first two interviews by two researchers (HS, MV). This has been added to the manuscript on p6:

“The interviews were conducted by three postdoctoral dementia researchers (one male (BT), two females (MV, HS); all with ample experience with conducting interviews with vulnerable populations), and two female medical students (EvdB, BvL). The medical students received a short interview training and got elaborate feedback on their first two interviews by two researchers (HS, MV).”

4. How do you ensure results rigor?

We described how we ensured results rigor on p6, the last paragraph in Methods. To clarify, we have now added more information about the quality control of the results. It now reads:

“To ensure inter-rater agreement, two researchers (HS, MV) did the analysis; similar and related codes were organized into higher level categories, which were restructured into a hierarchy of more abstract, overarching thematic clusters and sub-themes to form a coherent whole. The researchers engaged in a reiterant process of discussing areas of agreement and disagreement to enhance analytical rigor and achieve consensus. The research team was consulted if consensus could not be achieved. The outcomes were discussed within the research team and the other researchers provided feedback on the analysis. Qualitative analysis software ATLAS.ti was used to facilitate the analysis process.”

Results

1.I felt that family caregivers and health workers should be separated throughout. They are two different group of people.

We would like to thank the reviewer for their suggestion. We had considered separating the results for healthcare professional (HCP) and family carers, but decided only to do this when differences between the two groups appeared. If results for both groups were in line, separating them per group would result in repetition, increasing the length of the manuscript and reducing the readability. We have made adjustments throughout the manuscript to make differences in views and opinions between both groups more apparent.

2. The four themes: 1) impact on people with dementia, 2) impact on family carers, 3) impact on HCP, and 4) differences between home care and LTC could be renamed to reflect what they are. Now it looks like four aspects of enquiry, not themes. Were there any similar impacts for the 3 groups and 2 settings?

The four themes as described were the four main themes found during data analysis. Several subthemes were found within these four themes and are discussed under the subheadings. The four themes were not four aspects of enquiry, as is also reflected in the interview topic list (as found in the supplement). We now clarify this in the subheadings of the result section. This study did not show similar impacts for people with dementia, family carers and HCP. They experienced the social distance measures in their own way, as explained in the subheadings describing the themes. Similar impacts for the two settings are described throughout the results section and differences are highlighted under the subheading of the fourth theme “differences between home care and LTC”.

3. Table 1. Do they have increased or decreased care hours during the pandemic period?

We have added to Table 1 whether the participants experienced an increase, decrease or no change in the number of hours a week they spend on providing care for someone with dementia. Unfortunately, this question was not asked or answered by more than half of the family carers in the community setting. This has been added to the discussion as a limitation (p14).

4. You mentioned “In both home care and LTC, HCP’s workload increased…” As you mentioned earlier, many services were shut down. Why all workers’ workload increased? How about work hours?

We did not ask HCP about the exact number of work hours before and during the COVID-19 pandemic. However, we did ask HCP if the hours a week they provided care for persons with dementia during the COVID-19 pandemic was more, even, or less than before the pandemic. This has been added to Table 1 (p7-8).

Under ‘Impact on HCP’ in the Results section, we have described why HCP’s workload increased. In short, it was not just the extra hours they had to work. The workload also increased due to the additional responsibilities and care tasks, but also the changed and challenging circumstances in which they had to perform their tasks.  More information about the reasons HCP mentioned for the experienced increase in workload have been added to p10:

“For example, when a ward or individual resident needed to be quarantined, being more vigilant for COVID-19 symptoms, the use of PPE increased, and caring for COVID-19 patients intensified. HCP often had to work extra hours because of staff shortage due to sick leave. HCP in home care also indicated that it was burdensome having to decided what was essential care and having to communicate to patients when (part of) their care had to be temporarily stopped.”

Discussion

1. Please be more specific about practice suggestions. Is there any better way than restrictive measures? How can we do better? Please distinguish what has been done what has not been done.

Practice suggestions are now made on various points throughout the Discussion and Conclusion;

- p12 “There is now international consensus that nursing homes need to keep open for visitors safely with appropriate consideration of community transmission, nursing home outbreak status, and the preferences of residents, families, and HCP[7].”

- p12-13: “In order to preserve and protect the emotional wellbeing of people with dementia, social distancing measures should be replaced by provisions allowing social contact without significantly increasing risks of infections, and meaningful activities should be offered as much as possible, also in the community setting. Safe visiting practices as suggested by earlier research include visits in well ventilated separated areas or outdoors, proper infection control procedures, registration and screening of visitors, limiting the number, frequency and length of visits, and supervised by staff[6].”

- p13: “In general, we want to encourage family carers and HCP to practice physical distancing while staying socially engaged with people with dementia, as meaningful interactions and activities are vital for quality of life and remaining healthy. More effort could be made towards alternative ways of social contact, such as better facilitated distanced visiting, or finding other ways to keep family carers involved. Clear reasoning and communication with people with dementia, family carers and HCP about how social distancing measures are developed is vital, as is flexibility in enforcing those measures[6,7].”

-p13: “Pro-actively supporting the wellbeing of HCP is needed, as they have been exposed to high-stress environments, which may cause mental health issues[6,18].”

- p14: “In responding to the pandemic, it is essential to be mindful of the challenges that restrictive measures are creating for people with dementia and their family carers and to address these challenges head on. Therefore, we advise family carers and HCP to follow social distancing measurements while staying socially engaged with people with dementia. Also, we recommend to involve persons with dementia, their advocates, families, and HCP in discussions around restrictive measures, in particular regarding safe visiting.”

2. People staying home much longer in lockdown. If patients with dementia lived with family members, did they spend more time or have more interaction with them?

We did not specifically asked the family carers of community-dwelling persons with dementia if they interacted and spend more time with their relative. We had data of 4 family carers about whether they spend more, the same or less hours a week on caregiving tasks during the pandemic compared to before the pandemic. This has been added to Table 1.

Family carers of community-dwelling people with dementia indicated that care routines and responsibilities often intensified (due to the downscaling of home care services), suggesting more time spend with their relative (p9). Also, options for respite care stopped, so family carers who lived with the person with dementia spent even more time together. Respite care is important for family carers, providing options to relax, take a step back, and recharge their battery. Spending more time together in that case is not always necessarily best.

To speculate at bit more, care routines and responsibilities require the family carer to spend time with their relative, it may be a different experience for both the provider and receiver than when engaging together in a fun or meaningful activity together. It would be interesting to further investigate the impact of the COVID-19 measures on meaningful, fun or leisure activities for people with dementia and their family carers. Having fun, positive caregiving experiences may buffer to ameliorate the stress of caregiving.

3. Conclusion could be modified to be more related to your results.

We like to thank the reviewer for their suggestion. The conclusion of the Abstract has been adjusted to: “The social distancing  measures had a negative impact on persons with dementia, their family carers, and healthcare professionals. More attention is needed for community-dwelling people with dementia and family carers in times of social isolation.”

The Conclusion section has been revised thoroughly (p14), and now states:

“At the outbreak of the COVID-19 pandemic, dementia care policies had to be developed swiftly, and were most applicable to LTC residents. However, this study showed that needs of those involved may vary and that home care services required different approaches than LTC. More care during times of social distancing measures isolation is needed for community-dwelling people with dementia, especially those with a limited social network, or those in need of intensified care for other reasons. In responding to the pandemic, it is essential to be mindful of the challenges that restrictive measures are creating for people with dementia and their family carers and to address these challenges head on.A better balance between physical safety and well-being is necessary, as social isolation is a serious health threat for older people. Therefore, we advise family carers and HCP to follow social distancing isolation measurements while staying socially engaged with people with dementia. Also, we recommend to involve persons with dementia, their advocates, families, and HCP in discussions around restrictive measures, in particular regarding safe visiting.”

Reviewer 4 Report

A relevant topic - and nice results. I need more information on whether the home-dwelling people with dementia lived alone or together with family/spouses etc. - and reflections on this.

A shame that the voices of people with dementia could not be heard themselves - but on the other hand you explains very well the reasons for this.

Would it have been possible to make online interviews with the target group living in the community?

Author Response

Reviewer report 4

A relevant topic - and nice results. I need more information on whether the home-dwelling people with dementia lived alone or together with family/spouses etc. - and reflections on this.

We thank the reviewer for their comment. Of the family carers of a home-dwelling person with dementia, 70% were the partner of the person with dementia and living together with the care recipient. The other 30% (n=3) were the daughter of the person with dementia. Two provided daily care for their mother together with their father. Both daughters suffered a burn out due to their caregiving tasks, indicating they were heavily involved in the care tasks. The third daughter provided care for her father together with her two siblings. Her father lived alone in an assisted lived apartment for seniors, but she provided care at least every other day. These conditions made that they were deeply affected by the downscaling of home care services.

We have added more information about this throughout the manuscript:

- Additional information about the family carers of community-dwelling persons with dementia have been added on p7 (first paragraph under Results): “The family carers were either the partner (65%, n=13) or child daughter of the person with dementia (35%, n=7). Of the family carers of home-dwelling persons with dementia, the partners lived with the care recipient, two daughters provided daily care together with the partner of the care recipient, and one provided care together with her siblings to her parent who was living independently in an assisted living apartment for seniors.”

- p9 Results: “In the community, the ‘bubble’ around the person with dementia was kept small. It was mostly limited to visits from the primary family carer(s) if the person with dementia was living by themselves or only a handful of other family carers if the person with dementia lived with their partner.”

- p13 Discussion:

“Almost all family carers of community-dwelling people with dementia in our sample were either living with the person with dementia or providing daily.”

A shame that the voices of people with dementia could not be heard themselves - but on the other hand you explains very well the reasons for this.

Would it have been possible to make online interviews with the target group living in the community?

We would like to thank the reviewer for their interesting question. We believe it is not impossible to conduct interviews with community-dwelling persons with dementia during times of social distancing measures, but additional assistance would be needed. For example, we experienced that conducting online interviewer with some of the older family carers were sometimes challenging if they had no previous experience with using online communication applications. We would therefore suggest to use the telephone instead of online programs or to have a family member of HCP present – if the restrictive measures allow this - to help them get started and assist should any technical difficulties occur. The interview questions should also be slightly adapted, for example ask more about the here-and-now and about shorter time periods, to minimize feelings of failure.